# An Analysis Graph for Statistical Genetics Agents

Stephen Dorn*
University of California, Los Angeles
Los Angeles, CA, USA

Jason Mohoney*
Massachusetts Institute of Technology
Cambridge, MA, USA

Tim Kraska
Massachusetts Institute of Technology
Cambridge, MA, USA

Sam Madden
Massachusetts Institute of Technology
Cambridge, MA, USA

Noah Zaitlen
University of California, Los Angeles
Los Angeles, CA, USA

Margaux Hujoel
University of California, Los Angeles
Los Angeles, CA, USA

## ABSTRACT

Statistical genetics workflows have grown more complex in recent years, with new methods, software, genome annotations, and publicly available GWAS summary statistics outpacing what any one researcher can keep track of. The problem is sharpest for clinical investigators, who often have deep disease expertise but limited bandwidth to track an expanding post-GWAS toolkit. Coding agents can run the software, but command execution alone does not produce a reviewable, reusable analysis. We propose a graph-backed analysis agent for statistical genetics. Skills describe each analysis task; the agent executes them and uses the graph to pick methods, reconcile mismatched inputs, and route results between stages. The graph also records typed claims about every artifact, command, reference, and result, including dataset identity, genome build, linkage-disequilibrium (LD) reference panel, software version, quality-control summaries, and unresolved expert-review decisions. In a preliminary case study, the agent ran a nine-stage post-GWAS pipeline on a public inflammatory bowel disease (IBD) GWAS, finishing in 94.7 minutes on a single laptop. The agent reproduced the canonical IBD findings ($h^2$ = 0.148, IL23R R381Q PIP 0.97, immune-cell enrichment) and left a queryable, reviewable record of every method, reference, and decision.

**VLDB Workshop Reference Format:**
Stephen Dorn, Jason Mohoney, Tim Kraska, Sam Madden, Noah Zaitlen, and Margaux Hujoel. An Analysis Graph for Statistical Genetics Agents. VLDB 2026 Workshop: Biomedical Data Management Systems (BioDMS).

**VLDB Workshop Artifact Availability:**
The source code, data, and/or other artifacts have been made available at https://github.com/svdorn/statgen-skills.

## 1 INTRODUCTION

Public GWAS summary statistics [2], post-GWAS methods, and supporting reference data now expand faster than any one researcher can track. At the same time, a growing population of researchers arrive at genetic data from outside the methods community. Clinical investigators with deep biological expertise in a disease, such

*Equal contribution.

as gastroenterologists studying IBD or neurologists studying ALS, increasingly want to interpret association results themselves rather than depend on a methods collaborator for every preliminary analysis. They have data and biological intuition; what they lack is a tractable path through the post-GWAS toolkit.

That toolkit is large. Interpreting new genetic associations now involves numerous downstream analytical tasks such as SNP heritability estimation [1], stratified or cell-type heritability partitioning [7], transcriptome-wide association [9], polygenic score construction [13, 18], locus-level fine-mapping [17], cross-trait genetic correlation, and downstream pathway and gene-set analysis. Each tool depends on its own reference data, input format, and assumptions. A small mismatch in build, ancestry, or sample-size encoding can yield an incorrect result. Coding agents can do much of the mechanical work: reading documentation, installing software, formatting inputs, and executing commands. They cannot, on their own, record the run in a form a domain expert can later audit or reuse.

We represent statistical-genetics analysis state as a queryable, reviewable data object. A graph records durable analysis objects (dataset identity, genome build, LD reference panel, software version, command provenance, quality-control summaries, results, and unresolved expert-review decisions) and the typed claims that link them. Skills describe how to perform each analysis task, the agent executes them, and the graph records what each step used and produced. We motivate the design with two use cases (Section 2), describe the graph and skill interface (Section 3), and demonstrate the system on a public IBD GWAS (Section 4) running a nine-stage pipeline in under two hours on a single laptop that recovered the canonical IBD genetic architecture. Our contribution is a provenance layer for analysis validity that an agent and a human reviewer can query interchangeably, demonstrated through a 94.7-minute end-to-end pipeline whose steps remain available for later audit.

## 2 BACKGROUND AND USE CASES

Two recurring tasks motivate the analysis graph, both common in the working life of an investigator and both requiring the same orchestration layer.

### 2.1 Use Case 1: GWAS Interpretation

Genome-wide association studies (GWAS) identify genomic variants statistically associated with a disease or trait. Over the past 20 years, GWAS have been run on almost every disease or trait imaginable and have identified thousands of associations [16]. We can now discover variants faster than we can interpret them; the field

has shifted toward interpretation methods [8]. Investigators still construct new GWAS, either with larger sample sizes to increase power or for rarer diseases that only now have sufficient samples with linked genetic data.

When an investigator runs a new GWAS, they must run a battery of analysis tools to interpret what the associated loci mean biologically. Common follow-up analyses include mapping variants to genes and regulatory elements, identifying enriched tissues and cell types, fine-mapping to prioritize likely causal variants, computing and interpreting polygenic risk scores, and comparing to related traits via genetic correlation. Each analysis class has tens to hundreds of methods with corresponding software, and each step has its own reference data and required domain expertise.

These constraints are routine for trained statistical geneticists but difficult for investigators without specialized methods training. Our system aims to support clinicians and biologists in conducting preliminary, reviewable analyses while preserving decisions that require expert review; it can also accelerate the same analyses for trained geneticists.

### 2.2 Use Case 2: Testing a hypothesis with data

An investigator hears a conference talk or reads a recent paper that motivates a specific genetic hypothesis. For example, a recent report of somatic mutations associated with neurodegeneration in ALS patients [19] raises a question for investigators studying ALS: does somatic structural variation in mismatch-repair pathways contribute to ALS progression in C9orf72 repeat carriers, analogous to the role of somatic HTT-repeat instability in Huntington's disease [10]? The investigator can ask: does existing GWAS data support a related claim? Do low p-values cluster near mismatch-repair genes? Are annotations for DNA repair or repeat instability enriched in the ALS-association signal? Use Case 2 inverts Use Case 1: the investigator arrives with a question needing data, not data needing follow-up. The agent must therefore translate the hypothesis into genes, annotations, or variants; identify public datasets that can answer the questions; select methods matched to the data and question; and return evidence drawn from analysis of real data rather than from a literature summary.

*Related systems.* Graph execution traces are well established: CWL and BioCompute Objects record run provenance [3, 11], while BioCypher constructs typed biomedical knowledge graphs [15]. We target the layer between them: versioned, queryable claims about domain validity, including genome build, ancestry, LD panel, sample-size encoding, trait prevalence, variant identifiers, and unresolved expert decisions.

## 3 METHODS: THE ANALYSIS GRAPH

The system has three components: a library of *skills* that each document how to perform one statistical genetics task, an *agent* that reads a skill and executes the steps it prescribes, and a *graph* that records the objects each executed step consumed and produced. Figure 1 shows the architecture. The graph does not store bulk statistical data: summary-statistics files, reference panels, and tool outputs remain on disk and are referenced by checksum from the corresponding artifact nodes.

*Implementation.* The prototype used Claude Code with Opus 4.8 via Anthropic-hosted inference and a PostgreSQL graph exposed over MCP. The 94.7 minutes cover the end-to-end run excluding one-time reference downloads; the design is framework- and model-agnostic.

### 3.1 Data model

The graph is a directed, typed graph over a fixed schema. The core execution subgraph used to record an analysis has eight node types, shown in Figure 2: dataset, artifact, reference, method, software, command, result, and review. Each type has a fixed attribute schema. A dataset node, for example, carries genome build, ancestry composition, case and control counts, and a trait identifier; a result node carries a numeric value and its standard error. A broader catalog layer adds traits, genes, papers, and ontology terms; Figure 3 reports both layers.

Edges are typed and constrained to specific (source-type, target-type) pairs, shown in Figure 2: *exports* (dataset → artifact), *input to* (artifact → command), *used by* (reference → command), *ran* (command → software), *implements* (software → method), *produced* (command → result), and *has review* (result → review). An edge whose endpoint types are not paired in the schema is rejected at insert time.

### 3.2 Skills

A skill is a written recipe for one statistical genetics task. It names the tool to invoke (with its expected version and required reference data), the inputs the task takes, the outputs it produces, the quality-control checks to run on those outputs, and the graph nodes and edges the agent must write when the task completes successfully. Skills do not execute tools; the agent reads the skill, calls the tool, validates the outputs against the prescribed checks, and writes the prescribed nodes and edges to the graph.

### 3.3 Generations and queries

The graph is versioned as a sequence of immutable generations identified by monotonically increasing integers. A new generation is published whenever an ingestion pipeline or the agent commits new content; older generations remain readable. An agent session pins itself to one generation identifier on entry and reads against that identifier for its duration, so multi-step queries observe a single consistent state even while concurrent writers publish later generations.

The agent reads through a fixed catalog of parameterized graph queries that return node and edge identifiers rather than free text. The IBD case study in Section 4 exercises four query patterns: attribute lookup against a node identifier, typed-edge traversal between two nodes, attribute-predicate filtering over node sets, and provenance closure (the set of input artifacts and references on which a given result transitively depends).

## 4 PRELIMINARY EVALUATION

We evaluate the graph as a validity and orchestration layer for an end-to-end statistical genetics workflow. The case study follows Use Case 1: an investigator with a new GWAS asks for the full

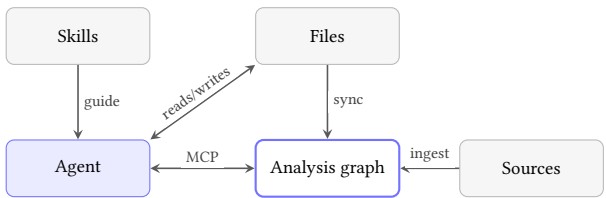

**Figure 1: Architecture. Skills guide the agent. The agent reads and writes files on disk, and reads and writes the analysis graph through an MCP interface. Files and source catalogs are synced into the graph.**

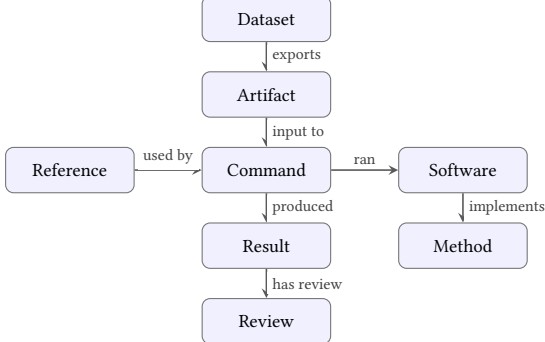

**Figure 2: Core analysis-execution subgraph: eight node types and the main typed edges between them. In the IBD case study these are instantiated as Dataset = GCST004131, Artifact = `ibd.munged.sumstats.gz`, Reference = 1000G EUR HapMap3 LD panel, Method = LD-score regression heritability, Software = `ldsc v1.0.1`, Command = `ldsc.py --h2 ...`, Result = $h^2 = 0.148$ (liability scale), Review = needs expert review.**

post-GWAS follow-up, with the graph mediating every tool choice, reference selection, and intermediate result.

## 4.1 Case study: IBD post-GWAS follow-up

We used the de Lange et al. 2017 inflammatory bowel disease GWAS [5] (accession GCST004131; 25,042 cases, 34,915 controls; European ancestry; GRCh37). A single agent session ran the nine-stage pipeline summarized in Table 1: sumstats fetch from GWAS catalog, Manhattan and QQ plots, LDSC munging and SNP-heritability estimation [1], stratified-LDSC tissue partitioning across 205 multi-tissue panels [7], TWAS FUSION [9] against GTEx v8 whole blood, polygenic-score construction with SBayesRC [18], fine-mapping of the chr1 *IL23R* locus with SuSiE [17], LocusZoom visualization, and a cross-trait $r_g$ calculation against twelve immune traits. End-to-end wall-clock time, including agent orchestration and local tool execution, was 94.7 minutes; first-run reference downloads of LD and genomic annotations (~6 GB) were one-time costs.

## 4.2 Biological findings

The pipeline reproduced the established IBD findings. Liability-scale heritability was $h^2 = 0.148$ (SE 0.013, intercept 1.15), and the only

**Table 1: Nine-stage IBD pipeline. All stages single-threaded on one macOS-arm64 laptop; reference downloads excluded.**

| # | Stage | Skill | Wall (min) |
|---|-------|-------|-----------|
| 1 | Fetch sumstats | `gwas-fetch` | 0.5 |
| 2 | Manhattan and QQ plots | `manhattan-qq` | 0.2 |
| 3a | LDSC munging | `ldsc munge` | 1.2 |
| 3b | SNP heritability | `ldsc h2` | 0.2 |
| 4 | S-LDSC, 205 tissues | `ldsc partitioned` | 18.1 |
| 5 | TWAS FUSION (whole blood) | `twas` | 9.0 |
| 6 | PRS, SBayesRC (3000 MCMC) | `prs` | 34.2 |
| 7 | Fine-map *IL23R* | `finemap region` | 1.3 |
| 8 | LocusZoom plot | `locuszoom` | 2.0 |
| 9 | Cross-trait $r_g$ (×12) | `ldsc rg` | 28.0 |
| **Total** | | | **94.7** |

GTEx tissue surviving Bonferroni correction across 205 candidates was whole blood ($p = 4.2 \times 10^{-7}$), consistent with IBD's immune origin. The top TWAS association was *CARD9* ($p = 5.1 \times 10^{-36}$), a canonical IBD gene reported in prior GWAS [12]. SuSiE fine-mapping of the most significant GWAS association at the chr1 *IL23R* locus identified rs11209026 (*IL23R* R381Q) with posterior inclusion probability 0.97, a canonical variant long established in the IBD literature [6]. Cross-trait genetic correlation showed the expected high correlation with Crohn's disease and ulcerative colitis, which are subtypes of IBD ($r_g \approx 0.9$), and significant associations with other immune traits such as ankylosing spondylitis ($r_g = 0.42$) and celiac disease ($r_g = 0.21$).

## 4.3 Decisions the graph informed

A code-only agent could in principle execute the same nine stages. In this run, the graph supplied recorded context at three points where mismatched inputs or implicit defaults could otherwise affect the result. A controlled comparison is future work.

*Reconciling mismatched inputs.* The cross-trait stage required pulling twelve comparator GWAS from disparate consortia whose ancestry labels used inconsistent vocabularies (European, EUR, and NR, annotated as European); the agent normalized these against the graph's controlled-vocabulary ancestry attribute and rejected candidates whose ancestry composition did not match the European discovery cohort, preventing biased $r_g$ estimates. The discovery sumstats and the SBayesRC LD reference for PRS were GRCh37 while the TWAS FUSION GTEx v8 weights were GRCh38; both builds were recorded as attributes on the dataset and reference nodes, so the agent detected the mismatch before launching TWAS and invoked the liftover skill to convert coordinates between builds. Because the sumstats lacked a per-SNP sample-size column, the agent read the case (25,042) and control (34,915) counts from the dataset node into the LDSC munging (QC) command. It also read IBD's population prevalence from the trait node to convert the heritability estimate to the liability scale [14], a step required for binary disease.

*Routing tools from prior results.* Several stages took parameters from earlier stages' results rather than from the user. The S-LDSC tissue scan identified whole blood as the only Bonferroni-significant GTEx tissue, and the agent used that recorded result as the active

TWAS reference panel, whereas a naive default would have run TWAS against every GTEx tissue, multiplying multiple-testing burden and diluting the immune signal. LDSC munging (QC) required a HapMap3 merge-alleles file and the correct signed-sumstats column name, both of which the agent read from the dataset's source manifest in the graph. Fine-mapping required variant IDs matching its LD reference panel; the panel's expected variant-ID format was recorded as a reference-node attribute, and the agent translated the sumstats rsIDs accordingly. In each case, the next command's arguments came from the prior tool's recorded result via the graph, not from the user.

*Choosing which software to run.* For the fine-mapping step, the agent used the graph to choose between methods. At the chr1 *IL23R* locus, two viable fine-mappers were available: SuSiE [17] and its infinitesimal extension SuSiE-inf [4]. The agent read the locus-level context recorded by earlier association and annotation passes and selected SuSiE-inf, which is designed to model diffuse background signal while retaining sparse large effects. In post-hoc comparison, SuSiE-inf produced 2 credible sets at the locus vs. 10 for SuSiE, more cleanly isolating the protective R381Q missense variant. A code-only agent would not have the necessary context to make this decision unless explicitly told to.

## 4.4 Graph construction on a laptop

To show that the graph builds quickly, we ingested a complete NHGRI-EBI GWAS Catalog [2] snapshot single-threaded on one macOS-arm64 laptop in 17.5 minutes (Table 2, Figure 3). The build produced 206,814 nodes and 755,168 edges and passed all 27 graph schema checks. The review queries used in the IBD case study (exact lookups for source artifacts, validity-path traversals from dataset to fine-mapped variant, ancestry-filtered comparator selection, and counts of trait-linked datasets) return in under 3 ms p95 against this snapshot.

**Table 2: Graph build on a laptop from a local GWAS Catalog copy. The graph stores metadata about catalog records, not the rows themselves.**

| Measurement | Value |
| --- | --- |
| Data source | GWAS Catalog snapshot |
| Build time | 17.5 min |
| GWAS Catalog rows read | 2.84M |
| Graph nodes | 206,814 |
| Graph edges | 755,168 |

## 5 DISCUSSION AND FUTURE WORK

Our prototype demonstrates a targeted proof of concept: a graph-backed agent can produce a more reviewable analysis record for a real statistical genetics workflow. In the IBD case study, the graph recorded every artifact, command, software version, reference asset, QC counter, result, and review state across nine pipeline stages, and the recorded claims were sufficient to reconstruct each tool choice and reference selection after the run.

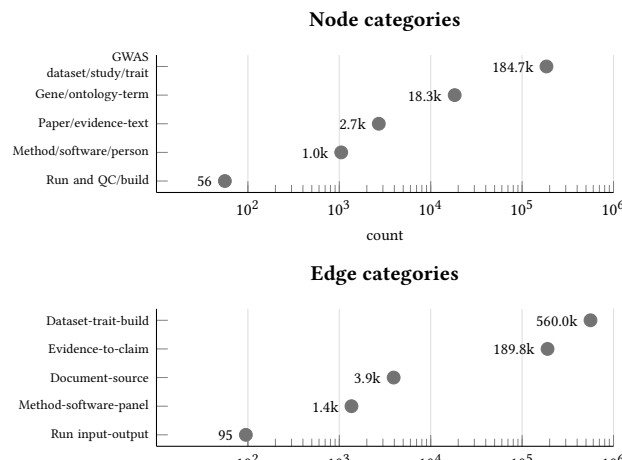

**Figure 3: Node and edge counts by category, log scale.**

*Limitations.* The graph can still be wrong in three ways: source metadata can go stale, build compatibility can be inferred too broadly, or a graph path can omit a statistical assumption a domain expert would supply by hand. The IBD run shows that this kind of knowledge can be encoded when it is in the schema: the graph correctly applied the LDSC liability-scale conversion using IBD's population prevalence [14]. However, additional analyses will require domain knowledge and reference data the graph does not yet encode.

*Improved evaluation.* We will compare the IBD case study across three configurations: code execution only, skills without the graph, and the full system. A blinded statistical geneticist will score correctness, compatibility reasoning, unsupported claims, and reviewer usefulness. We will also test whether Use Case 2 produces defensible analyses from free-text hypotheses and measure usability and expert-review time in a user study.

*Generalization.* The architecture is not specific to statistical genetics. Any analytic domain whose validity depends on cross-tool compatibility (e.g., single-cell omics with reference atlases and batch corrections) faces the same problem. Future work will measure whether graph-backed reviewability reduces expert time and prevents validity errors on workflows where the answer is not already known.

## AUTHORS

**Stephen Dorn** (biomedical community). PhD student in Biomathematics at UCLA's Department of Computational Medicine. Develops statistical and computational methods to better understand genetic influences on human diseases and complex traits.

**Jason Mohoney** (data management community). Postdoctoral researcher in the Data Systems Group at MIT CSAIL. Studies AI systems and their scientific applications.

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
