# OpenReview forum: "An Analysis Graph for Statistical Genetics Agents"
_VLDB.org/2026/Workshop/BioDMS — BioDMS 2026 ProjectTalk_

### Official Review · Reviewer_VP1k · 2026-06-10

**Summary:**

The paper proposes a graph-backed analysis agent for post-GWAS statistical genetics.
The system combines written recipes for individual analysis tasks), an LLM agent that executes them, and a versioned analysis graph that records typed claims about every artifact and decision.
The graph's purpose is "analysis-validity provenance".
In a preliminary case study, the agent ran a nine-stage post-GWAS pipeline on a public IBD GWAS in 94.7 minutes on a laptop.

**Confidence Of Review:**

2

**Detailed Feedback Points:**

1. The distinction between execution provenance and analysis-validity provenance well-positioned against workflow systems and biomedical KGs.
2. Recovering well-established IBD genetics end-to-end on a laptop shows concrete validation of the proposed system.
3. Versioned graph analysis is quite interesting for reproducibility. Artifacts are publicly available.

**Relevance For Biodms:**

3

---

### Official Review · Reviewer_MBjv · 2026-06-16

**Summary:**

The Project Talk submission “An Analysis Graph for Statistical Genetics Agents” proposes an orchestration system for agents performing statistical analysis of genetic data. The core piece of the orchestration logic is an analysis graph that serves to both constrain for the agent and explain the analysis steps in a structured format. The potential capabilities of the system are illustrated with two case studies.

**Confidence Of Review:**

3

**Detailed Feedback Points:**

Strengths:
1. The core idea of building a structured analysis graph to help guide the analysis and review the results is an interesting approach that is worth further exploration and development.

Suggested Improvements:
1. The use of existing technology should be clarified. Is the system making API calls to an LLM? Is it building on top of existing orchestration tools like Claude Code? Is it running a local model? In particular the description of the use case running on a laptop CPU made me question where the LLM inference was running.
2. The paper suggests that the current system can replace the need for a methods collaborator. I find the current phrasing to be a little too strong and could be toned down or hedged more.

**Relevance For Biodms:**

3

---

### Official Review · Reviewer_7dYg · 2026-06-16

**Summary:**

The paper presents a graph-based analysis decomposition and tracing system enabling domain experts to work with agentic data analysis of genomic data.

**Confidence Of Review:**

3

**Detailed Feedback Points:**

S1 The solution of the analysis graph system that captures the executed “skills” (traces?), executed versions, and annotations/reviews, is well motivated by two use-cases, on interpretation and hypothesis analysis.

S2 The effectiveness of the system is illustrated by examples of insights/issues that are surfaced through the system, which are not surfaced through an agent-based system that doesn’t capture traces/versions. The paper also presents an efficiency analysis.

S3 The paper fits the venue well as it has some unsolved limitations, such as incorrectness, evaluation, and generalizability. This could be interesting to discuss with the audience and has potential to yield effective collaboration / follow-up work.

W1 The system’s main goal is making agent-based analysis of healthcare data more reviewable and reusable for domain experts. To this end, it would be interesting to conduct a user study to understand the usability of the design.

W2 Decomposing agentic traces into a (graph) structure of the operations is quite common and it would be interesting to see what the specific challenges/features are that this system needs to address for biomedical data. A study of related work in this area seems useful.

**Relevance For Biodms:**

2